# CB2 Agonist GW842166x Protected against 6-OHDA-Induced Anxiogenic- and Depressive-Related Behaviors in Mice

**DOI:** 10.3390/biomedicines10081776

**Published:** 2022-07-22

**Authors:** Xiaojie Liu, Hao Yu, Bixuan Chen, Vladislav Friedman, Lianwei Mu, Thomas J. Kelly, Gonzalo Ruiz-Pérez, Li Zhao, Xiaowen Bai, Cecilia J. Hillard, Qing-song Liu

**Affiliations:** 1Department of Pharmacology and Toxicology, Medical College of Wisconsin, 8701 Watertown Plank Road, Milwaukee, WI 53226, USA; xiaojieliu@mcw.edu (X.L.); haoyu@mcw.edu (H.Y.); bchen@mcw.edu (B.C.); vfriedman@mcw.edu (V.F.); lmu@mcw.edu (L.M.); tjkelly@mcw.edu (T.J.K.); gperez@mcw.edu (G.R.-P.); chillard@mcw.edu (C.J.H.); 2Department of Exercise Physiology, Beijing Sport University, Beijing 100084, China; zhaolispring@126.com; 3Department of Cell Biology, Neurobiology and Anatomy, Medical College of Wisconsin, 8701 Watertown Plank Road, Milwaukee, WI 53226, USA; xibai@mcw.edu

**Keywords:** CB2 agonist, GW842166x, 6-OHDA, dopamine, anxiety and depression

## Abstract

In addition to motor dysfunction, patients with Parkinson’s disease (PD) are often affected by neuropsychiatric disorders, such as anxiety and depression. In animal models, activation of the endocannabinoid (eCB) system produces anxiolytic and antidepressant-like behavioral effects. CB2 agonists have demonstrated neuroprotective effects against neurotoxin-induced dopamine neuron loss and deficits in motor function. However, it remains unknown whether CB2 agonism ameliorates anxiogenic- and depressive-like behaviors in PD models. Here, we report that the selective CB2 agonist GW842166x exerted neuroprotective effects against 6-hydroxydopamine (6-OHDA)-induced loss of dopaminergic terminals and dopamine release in the striatum, which were blocked by the CB2 antagonist AM630. We found that 6-OHDA-treated mice exhibited anxiogenic- and depressive-like behaviors in the open-field, sucrose preference, novelty-suppressed feeding, marble burying, and forced swim tests but did not show significant changes in the elevated plus-maze and light–dark box test. GW842166x treatments ameliorated 6-OHDA-induced anxiogenic- and depressive-like behaviors, but the effects were blocked by CB2 antagonism, suggesting a CB2-dependent mechanism. These results suggest that the CB2 agonist GW842166x not only reduces 6-OHDA-induced motor function deficits but also anxiogenic- and depressive-like behaviors in 6-OHDA mouse models of PD.

## 1. Introduction

Parkinson’s disease (PD) is caused by the progressive degeneration of dopamine neurons in the substantia nigra pars compacta (SNc) [1,2]. When the degeneration of dopamine neurons reaches the threshold of an ~80% reduction in dopamine release within the striatum, PD patients begin to exhibit classical motor symptoms, including bradykinesia (slowness of movement), movement rigidity, and resting tremors [2,3,4]. Although these motor symptoms are useful in diagnosing PD, patients often demonstrate additional non-motor symptoms, such as hyposmia (a decreased sense of smell), sleep disturbances, and co-morbid psychiatric disorders [5,6]. Among these comorbid disorders, anxiety and depression are the most common, with approximately 40–50% of PD patients also being diagnosed with anxiety or depression [7,8,9]. In addition, the rate of severe depression among those with PD is twice as high as seen in other equivalently disabled patients [10,11]. The occurrence of depression has been identified as the greatest predictor of reduced quality of life in PD patients [12]. Although many treatments have been developed to alleviate motor dysfunction in those affected by PD, less attention has been directed toward the treatment of anxiety, depression, and other non-motor symptoms.

Accumulating evidence indicates that the endocannabinoid (eCB) system is a valuable target for neuroprotection against PD [13]. JZL184, a selective inhibitor of monoacylglycerol lipase (MAGL), an enzyme that degrades eCB 2-arachidonoylglycerol (2-AG) [14,15], demonstrates neuroprotective effects against neurodegeneration in mouse models of Alzheimer’s disease [16] and PD [17]. Prior research has shown that non-selective CB1/CB2 agonists reduced the neurotoxin-induced loss of dopamine neurons [18,19,20] and alleviated motor function deficits [19,21,22]. In phase II clinical trials, the partial CB1/CB2 agonist nabilone ameliorated anxiety and sleep disturbances [23]. As a result, there has been an intense interest in harnessing the eCB system for the pharmacotherapy of depression [24]. Interestingly, CB1 receptor polymorphism has been linked to a reduced prevalence of depression in PD patients but not in control subjects [25]. We have shown that the MAGL inhibitor JZL184 produces anxiolytic and antidepressant-like behavioral effects in a chronic stress model of depression, and that these effects are mediated by the activation of CB1 receptors [26,27]. However, the usage of CB1 agonists may lead to substance abuse and dependence, including tolerance and withdrawal [28], which might limit their therapeutic potential.

Selective CB2 agonists lack psychoactivity [29] and may be beneficial for alleviating both motor and non-motor deficits in PD. CB2 mRNA and protein are expressed in midbrain dopamine neurons [30,31,32]. Expression of the CB2 gene is significantly elevated in the SNc of postmortem brains of PD patients [33]. We, and others, have found that CB2 agonists demonstrate neuroprotective effects against neurotoxin-induced dopamine neuron loss and deficits in motor function [34,35]. As the CB2 agonist GW842166x exhibited good bioavailability and was found to be safe and well-tolerated by patients without serious adverse effects [36], we employed a neurotoxic model of PD through unilateral injection of 6-hydroxydopamine (6-OHDA) into the striatum and examined the neuroprotective effects of treatment with the CB2 agonist GW842166x [37]. We found that GW842166x reduced 6-OHDA-induced dopamine neuron loss in the SNc and alleviated motor behavior deficits assessed in the balance beam, pole, grip strength, rotarod, and amphetamine-induced rotation tests [37]. However, it was unclear whether CB2 activation with GW842166x also improves non-motor behavioral deficits in the 6-OHDA model of PD.

The present study determines whether GW842166x exerts neuroprotective effects against 6-OHDA-induced loss of dopaminergic axonal terminals and dopamine release in vitro and in vivo. Neurotoxin-induced rodent models of PD exhibit depressive-like behaviors [38]. We also examined whether 6-OHDA-induced lesions of dopamine neurons causes anxiogenic- and depressive-like behavioral phenotypes and whether the CB2 agonist produces anxiolytic and anti-depressive-like effects. Together with prior studies [19,21,22], our results indicate that the CB2 agonist improves both motor and non-motor deficits in the 6-OHDA model of PD.

## 2. Materials and Methods

### 2.1. Animals

C57BL/6J mice (10–12 weeks old) were acquired from The Jackson Laboratory (Jax stock#: 000664, Bar Harbor, ME, USA). Approximately equal numbers of male and female mice were utilized in all experimental groups. Mice were housed 4–5 per cage under a 14 h light/10 h dark cycle with food and water available ad libitum, unless stated otherwise. Temperature and humidity were maintained at 23 ± 1 °C and between 40 and 60%, respectively. All animal maintenance and use protocols were evaluated and approved by the Institutional Animal Care and Use Committee of the Medical College of Wisconsin. Four cohorts of mice were used. In Experiment 1 (Figure 1), tyrosine hydroxylase (TH) and NeuN immunofluorescent staining was performed three weeks after intra-striatum 6-OHDA injection (*n* = 3 mice). In Experiment 2 (Figures 2 and 3), mice received stereotaxic surgeries for intra-striatal 6-OHDA or control injections, followed by three weeks of drug treatment. Immunohistochemical TH staining of the striatum and fast-scan cyclic voltammetry (FSCV) recordings of dopamine dynamics in striatal slices were performed after drug treatment (*n* = 28 mice). In Experiment 3 (Figure 4), mice received stereotaxic intra-striatal injection of 6-OHDA (or control) and dopamine sensor (see Section 2.2), followed by three weeks of drug treatment, and fiber photometry recording (*n* = 27 mice). In Experiment 4 (Figures 5 and 6), mice received stereotaxic surgeries and three weeks of drug treatment, and anxiogenic- and depressive-like behavioral tests were performed (*n* = 35 mice).

### 2.2. Stereotaxic Surgeries

Intra-striatal 6-OHDA injection was performed as described previously [37]. First, mice received desipramine treatment (25 mg/kg, i.p.; Sigma-Aldrich, St. Louis, MO, USA) to inhibit 6-OHDA uptake by noradrenergic neurons [39]. After 30 min, anesthesia was performed with ketamine (90 mg/kg, i.p.) and xylazine (10 mg/kg, i.p.). Mice were then placed in a robot stereotaxic system (Neurostar, Tübingen, Germany), and 2 µL of 6-OHDA (2 µg/µL in PBS with 0.02% sodium L-ascorbate) or control (PBS with 0.02% sodium L-ascorbate) was administered (60 nl/min over 5 min) to the striatum at AP: +0.5 mm, ML: ±1.8 mm, DV: −3.0 and –2.0 mm [40] through a Nanoject III Programmable Nanoliter Injector (Drummond Scientific Company, Broomall, PA, USA).

For fiber photometry, mice first received an intra-striatal injection of 6-OHDA or control. Next, 200 nl of a 1:1 AAV9-hSyn-GRAB_DA2m_ and AAV8-hSyn-mCherry mixture was injected at the same site. A low autofluorescence fiber-optic cannula (OD = 200 µm, NA = 0.57; Doric Lenses, Inc. Québec, QC, Canada.) was implanted in the striatum above the site of AAV injection. The cannula was secured to the skull by two 1.6 mm screws and C&B Metabond cement. Following completion of all surgery, mice received an analgesic (buprenorphine-SR, 0.05 mg/kg, s.c.), and were fed liquid food (20% Ensure^®^ Original Vanilla Nutrition Powder; Abbott Laboratories, Abbott Park, IL, USA) for one week.

### 2.3. Drug Treatment

Mice began a three-week drug treatment regimen starting the day after stereotaxic surgeries. Four essential groups were utilized as described previously [37]. These groups were devised to determine whether GW842166x was effective in reducing 6-OHDA induced loss of dopaminergic terminals and the associated increases in anxiogenic- and depressive- like behaviors. Group 1: Control followed by daily i.p. vehicle (35% PEG 200 + 10% cremophor EL + 55% sterile saline) injections; Group 2: 6-OHDA followed by daily i.p. vehicle injections; Group 3: 6-OHDA followed by daily i.p. injections of 1 mg/kg GW842166x; Group 4: 6-OHDA followed by daily i.p. injections of 1 mg/kg GW842166x + 10 mg/kg AM630. After completion of drug treatments, immunohistochemical staining, fast-scan cyclic voltammetry (FSCV), fiber photometry, or behavioral tests were performed.

### 2.4. Immunohistochemistry (IHC)

Anesthesia, transcardial perfusion, brain fixation, and slice cutting were performed as previously described [37]. All sections were incubated with primary antibody at 4 °C for 48 h. For striatal sections, the primary antibody targeted tyrosine hydroxylase (TH, rabbit, 1:300, Santa Cruz Biotechnology, Inc., Dallas, TX, USA; SC-14007), and for midbrain sections the primary antibody targeted TH (rabbit, 1:300, Santa Cruz Biotechnology, Inc., Dallas, TX, USA, SC-14007) and NeuN (mouse, 1:300, Cell Signaling Technology, Inc., Danvers, MA, USA, #94403s). All sections were rinsed three times in PBS for 15 min each, then incubated in secondary antibodies for 4 h at room temperature. Midbrain sections were incubated in anti-rabbit Alexa-555 (1:300, Cell Signaling Technology, Inc., Danvers, MA, USA, #4413) and anti-mouse Alexa-488 (1:300, Cell Signaling Technology, Inc., Danvers, MA, USA, #4408) and striatal sections were incubated in anti-rabbit IgG HRP-conjugated (1:200, Jackson ImmunoResearch Labs, West Grove, PA, USA, #170-6515) secondary antibody. Midbrain sections were imaged with a Leica SP8 Upright confocal microscope. In striatal sections, immunoreactivity was visualized by applying 3,3′-Diaminobenzidine (DAB) from a Substrate Kit (SK-4100; Vector Laboratories, Inc., Newark, CA, USA) for 5 min, then rinsed with PBS for 5 min to cease the reaction. Sections were rinsed in PBS a second time, and were then dehydrated, cover slipped, imaged with a Hamamatsu Slide Scanner and analyzed by ImageJ 1.53k (National Institutes of Health, Bethesda, MD, USA) [41]. Briefly, images were first converted to an 8-bit format. The region of interest was outlined, and the mean gray values of the striatum and cortex were measured. Then, the images were processed with background subtraction [42,43]. The optical densities were expressed as the percentage of the side contralateral to 6-OHDA or control solution injection.

### 2.5. Fast-Scan Cyclic Voltammetry (FSCV)

Mice were anesthetized by isoflurane inhalation and decapitated. The brain was removed, trimmed, and embedded in low-gelling-point agarose. Coronal striatal slices were prepared using a vibrating slicer (Leica VT1200s, Nussloch, Germany) as described previously [44]. Slices were cut in solution containing (in mM): 110 choline chloride, 26 NaHCO_3_, 1.25 NaH_2_PO_4_, 2.5 KCl, 0.5 CaCl_2_, 7 MgSO_4_, 11.6 sodium ascorbate, 3.1 sodium pyruvate, and 25 glucose. The slices were allowed to recover in ACSF for at least 30 min prior to recording. All solutions were continuously saturated with 95% O_2_ and 5% CO_2_.

A 7µm cylindrical carbon fiber microelectrode encased in glass (Goodfellow Corporation, Oakdale, PA, USA) was used to detect dopamine release with FSCV. The microelectrode (100–150 µm exposed final length of) was lowered into the striatum at approximately: AP, 0.9–1.7 mm; ML, ±1.8 mm; DV, −3 mm, as determined by visual inspection of the slice with a mouse brain atlas [40]. The microelectrode contained 150 mM KCl solution. Demon software [45,46] and a Chem-Clamp Potentiostat (Dagan Corporation, Minneapolis, MN, USA) were used to apply triangular waveforms (hold at −0.4 V, then increase to 1.3 V at 400 V/s). A stimulating electrode was placed ~100 µm from the carbon fiber microelectrode to provide a single electrical stimulus pulse (250 μA, 0.1 ms duration). The microelectrodes were calibrated with 0.1–1 µM dopamine in the ACSF after recordings. Recordings were performed at 32 ± 1 °C, maintained using an automatic temperature controller (Warner Instruments LLC, Hamden, CT, USA).

### 2.6. In Vivo Fiber Photometry to Detect Dopamine Release in the Striatum with GRAB_DA2m_

A fiber photometry console and low-power LED drivers (LEDD_4; Doric Lenses, Inc. Québec, QC, Canada) were used to power 465 and 560 nm LEDs. The intensities were modulated at nonsynchronous frequencies (572.21 and 1017 Hz for 465 and 560 nm, respectively) using a lock-in amplifier to filter emissions arising from overlapping reporter excitation spectra. Excitation signals were directed through a fluorescence MiniCube (FMC6_IE(400–410)_E(460–490)_F(500–550)_ E(555–570)_F(580–680)_S; Doric Lenses, Inc., Québec, QC, Canada) to a fiber-optic patch cable (MFP_200/230/900-0.57_1.0_FCM_ZF1.25_LAF; Doric Lenses, Inc., Québec, QC, Canada), and rotary joint (FRJ_200/230/LWMJ-0.57_1m_FCM_0.15_ FCM; Doric Lenses, Inc., Québec, QC, Canada) to a previously implanted fiber-optic cannula connected by a 1.25 mm mating sleeve (Sleeve_ZR_1.25-BK; Doric Lenses, Inc., Québec, QC, Canada). Fluorescence emissions from each reporter were returned to the MiniCube through the same patch cable and directed to independent photoreceivers (Model 2151, DC low setting; Newport Corporation, Irvine, CA, USA). Prior to recording, all patch cables were exposed to high intensity stimulation with 465 nm and 560 nm excitation light to further reduce autofluorescence, and LED driving power was adjusted to yield a power output of 20 µW prior to attachment to a mouse. After attachment, mice were habituated to an open field chamber for at least 15 min, and dopamine transients were recorded for 3 min. Doric Neuroscience Studio controlled data acquisition at a rate of 120 Hz (12 kHz, decimated by 100). Signals from photoreceivers were demodulated and low-pass filtered at 6 Hz by the software.

Fiber photometry data were analyzed using custom Python code. ΔF/F values over time were calculated by applying a linear fit to the data and then applying the equation: $\Delta \frac{F}{F}=\frac{fitted\;465\;\mathrm{nm}-fitted\;560\;\mathrm{nm}}{fitted\;560\;\mathrm{nm}}$. Mini-analysis software (Bluecell Co., Seoul, Korea) was used to analyze the ΔF/F values for dopamine event frequency. The onset of an event was defined by a z-score higher than the mean plus the standard deviation multiplied by 3 of the z-scores of recorded fluorescence during the prior 0.5 s. The resultant peak frequencies were compared across groups [47]. After completion of the experimentation, AAV expression and localization of optic fiber implantation in the striatum were verified by immunohistochemistry and microscopy.

### 2.7. Behavioral Tests

Beginning the day after the completion of the drug treatments, mice were subjected to a battery of tests of anxiogenic- and depressive-like behaviors (see timeline in Figure 5). Behavioral tests have been described in detail in our recent study [48]. Experimenters were blinded to the treatment of animals during the behavioral tests and analysis. Behavioral tests are listed below in the order they were performed, and only one behavioral test was conducted per day. Behavioral tests were performed between 9:00 AM and 12PM each day unless stated otherwise. Less stressful behavioral tests were performed before more stressful behavioral tests.

#### 2.7.1. Open Field Test (OFT)

Mice were placed individually in one corner of a custom-made open field chamber (45 cm length × 45 cm width × 30 cm depth) and allowed to freely explore the arena for 10 min. The locomotor activity was monitored and recorded using the ANY-maze automated video-tracking system (Stoelting Co., Wood Dale, IL, USA). Total distance traveled in the open field chamber and time spent in the center of the chamber were calculated. The center was defined as the central 22.5 cm × 22.5 cm area of the open field chamber.

#### 2.7.2. Sucrose Preference Test (SPT)

Mice were individually housed and trained to drink from two sipper tubes (Drinko Measurer, Amuza Inc., San Diego, CA, USA) filled with drinking water for one week. The Drinko Measurer prevents liquid leaks and allows for accurate measurements of drinking volume. Mice were deprived of food and water for 8 h prior to the SPT. During the 16-h test period, one sipper tube contained drinking water and the other contained 1% sucrose in drinking water. The consumption of sucrose solution and water during this period was measured. Sucrose preference (%) was calculated as sucrose solution consumed divided by the total amount of solution consumed.

#### 2.7.3. Light—Dark Box (LDB)

The LDB test was performed in an apparatus (46 cm length × 20 cm width × 20 cm depth) with two compartments. One compartment (30 cm length × 20 cm width × 20 cm depth) has white walls and is exposed to light and the other compartment (16 cm length × 20 cm width × 20 cm depth) has black walls and is covered, with an opening (6 cm tall × 5 cm wide) to allow for transit between the two compartments. Mice were first individually placed into the light side of the box and allowed to habituate for 10 min prior to location tracking with ANY-maze. The amount of time spent in the light side and the number of transits between light and dark sides were quantified.

#### 2.7.4. Elevated Plus Maze (EPM)

The EPM apparatus (Stoelting Co., Wood Dale, IL, USA) includes a central platform (5 cm × 5 cm) elevated to 40 cm and four arms which extend from each edge of the central square, consisting of two open arms (35 cm × 5 cm) that are perpendicular to two closed arms (35 cm × 5 cm × 15 cm). Mice were placed in the center platform facing a closed arm and allowed to freely explore the maze for 5 min while the position of the mice was tracked with ANY-maze. The time spent in open arms and the number of entries into open arms were quantified.

#### 2.7.5. Novelty-Suppressed Feeding (NSF)

Mice that had been food deprived for 24 h were put individually into one corner of a novel open field chamber (50 cm long × cm wide × 30 cm deep). In the center of the chamber, a round white filter paper (8 cm diameter) held 2–3 regular chow pellets, and the latency to feed was measured, as described previously [48]. The latency to feed in the home cage was measured immediately following testing in the novel environment.

#### 2.7.6. Marble Burying Test (MBT)

Twenty-four dark blue marbles (1.35 cm diameter) were placed evenly on the top of the bedding (7–8 cm deep) in animal home cages (30 cm length × 18 cm width × 12 cm depth). Mice were put individually into their home cage with no cage lid and allowed to freely explore for 20 min. Marbles were considered buried if at least two thirds of an individual marble was covered in bedding, and the number of marbles buried following a 20 min session were counted.

#### 2.7.7. Forced Swim Test (FST)

FST apparatus consisted of glass cylinders (13 cm diameter, 25 cm tall) filled with water (30 ± 1 °C) to a depth of ~18 cm. A mouse was placed into the apparatus and recorded with video for 6 min. The time the mouse spent immobile during the last 4 min was scored manually, as described in our previous study [48].

### 2.8. Chemicals

The sources of chemicals were the following: 6-OHDA, (Sigma-Aldrich, St. Louis, MO, USA); GW842166X and AM630, (Cayman Chemical Company, Ann Arbor, MI, USA. 35% PEG 200 (Electron Microscopy Sciences, Hatfield, PA, USA), 10% cremophor EL (Sigma-Aldrich, St. Louis, MO, USA), and 55% sterile saline with gentle heating was used as the vehicle for GW842166X and AM630 [49]. All other common chemicals were obtained from Sigma-Aldrich (St. Louis, MO, USA).

### 2.9. Statistics

All results are presented as the mean ± SEM. Data sets were compared with either Student’s *t*-test or one-way ANOVA followed by Tukey’s post hoc analysis using Origin 2022b (OriginLab Corp., Northampton, MA, USA). Post hoc analyses were performed only when ANOVA yielded a significant effect. Results were considered to be significant at *p* < 0.05.

## 3. Results

### 3.1. CB2 Activation Protected against the 6-OHDA-Induced Degeneration of Dopamine Axonal Terminals in the Striatum

The 6-OHDA model of PD relies on the fast-acting and selective neurotoxicity of 6-OHDA to dopamine neurons [50]. C57BL/6J mice received a single unilateral injection of 6-OHDA or control at two sites in the striatum (Figure 1A) and were euthanized for immunohistochemistry three weeks later. Staining for the dopaminergic marker (TH^+^) and pan-neuronal marker NeuN demonstrated that 6-OHDA selectively induced the degeneration of SNc dopamine neurons while sparing both non-dopamine SNc neurons and dopamine neurons in the neighboring VTA (Figure 1B).

**Figure 1 biomedicines-10-01776-f001:**
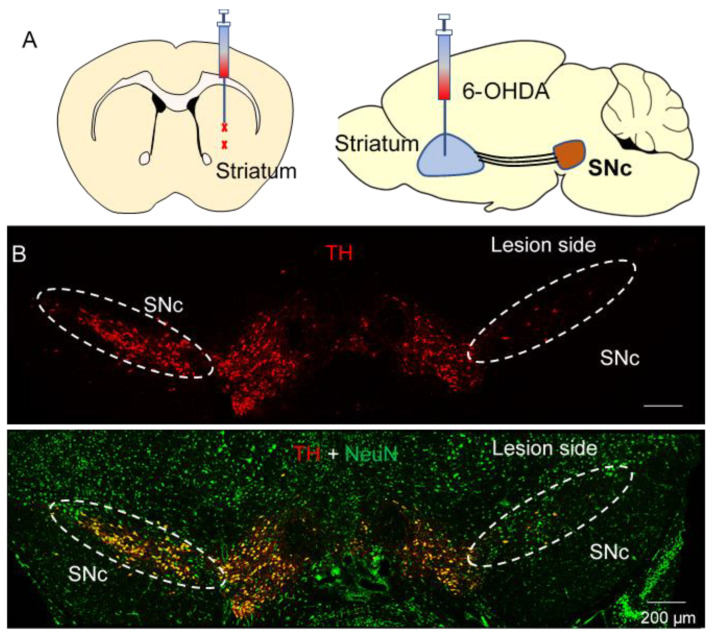
6-OHDA induced selective degeneration of SNc dopamine neurons. (**A**) Diagram showing the two targeted 6-OHDA injection sites in the striatum, where it is taken up by SNc axonal terminals. (**B**) Immunohistochemistry for TH in the SNc and neighboring VTA of mice that received unilateral injections of 6-OHDA (**top**). Immunohistochemistry for TH overlaid with NeuN shows the selectivity of 6-OHDA for dopamine neurons (**bottom**), *n* = 3 mice.

We have recently shown that the selective CB2 agonist GW842166x protects against 6-OHDA-induced dopamine neuron loss in the SNc [37]. Here, we examine whether GW842166x also protects against 6-OHDA-induced degeneration of dopaminergic terminals in the striatum. After unilateral injection of 6-OHDA into the striatum, mice began receiving the vehicle, GW842166x (1 mg/kg), or GW842166x (1 mg/kg) + AM630 (10 mg/kg) treatments daily for three weeks, the same as we have described previously [37]. Immunohistochemical staining for TH^+^ axonal terminals in the striatum was performed as described [37,51]. The experiment timeline is shown in Figure 2A. One-way ANOVA indicated significant effects of GW842166x treatments and 6-OHDA injection on the optical density of TH^+^ axonal terminals (*F*_3,23_ = 36.6, *p* < 0.001; Figure 2B,C). Tukey’s post hoc tests indicated that 6-OHDA reduced the optical density of TH^+^ axonal terminals in the striatum compared with the control (*p* < 0.001; Figure 2C). Treatments with GW842166x reduced degeneration of TH^+^ axonal terminals in the striatum (*p* = 0.002; Figure 2C) and these effects were blocked by co-treatment with the CB2 antagonist AM630 (*p* = 0.013; Figure 2C). Thus, GW842166x protected against the degeneration of dopamine axonal terminals following 6-OHDA injection, which resulted from the activation of CB2 receptors. There were no significant sex differences between treatment groups for any measure, and the data for males and females were pooled in this and subsequent experiments.

**Figure 2 biomedicines-10-01776-f002:**
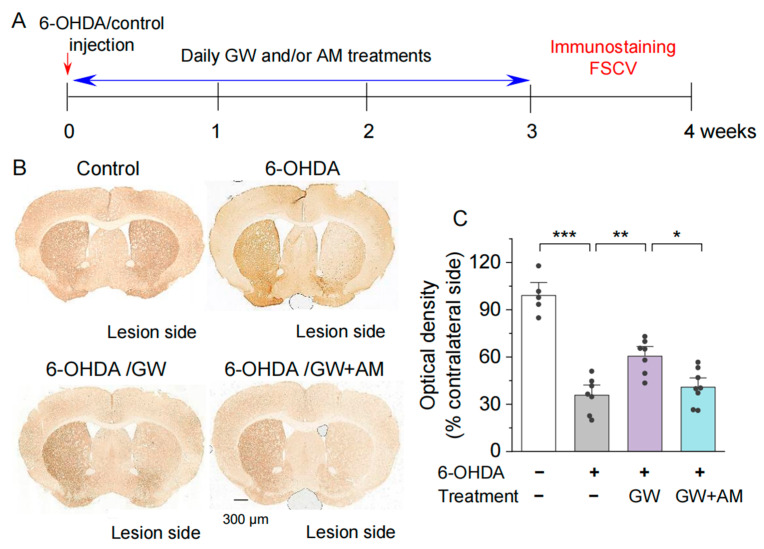
GW842166x (GW) was neuroprotective against 6-OHDA-induced loss of dopaminergic axonal terminals in the striatum. (**A**) Timeline of intra-striatal injection, drug treatments, and histology or FSCV recording. (**B**) DAB staining for TH+ axonal terminals in the striatum of mice that received intra-striatal injection of 6-OHDA or control solution and chronic GW or GW + AM630 (AM) treatments. (**C**) Summarized data showing that the optical density of TH+ axonal terminals in the striatum was significantly decreased in the 6-OHDA group relative to control (*** *p* < 0.001; *n* = 5–7 slices from 3–4 mice). Chronic GW treatments reduced 6-OHDA-induced loss of dopaminergic terminals (** *p* = 0.002; *n* = 7–7 slices from 4–4 mice), and this effect was blocked by AM co-treatment (* *p* = 0.013; *n* = 7–8 slices from 4–4 mice).

### 3.2. CB2 Agonism Protected against 6-OHDA-Induced Decrease in Striatal Dopamine Release In Vitro and In Vivo

Having shown that GW842166x protects against 6-OHDA-induced degeneration of dopamine cell bodies [37] and axonal terminals (Figure 2), we next examined these different treatments on dopamine release in the striatum. We used the FSCV technique [52] to detect dopamine release in response to electrical stimulation in striatal slices. Striatal slices were prepared from adult mice that received microinjections of 6-OHDA or the control and three weeks of drug treatments, the timeline of which is described in Figure 2A. FSCV recordings were blind to treatment conditions, 3–4 slices were used from each animal. A single pulse stimulation with a fixed intensity evoked a robust dopamine transient in striatal slices from control-injected, vehicle-treated mice. Dopamine transients were barely detectable in slices from 6-OHDA-injected mice that received vehicle treatment (*F*_3,36_ = 65.0, *p* < 0.001; Figure 3A,B). However, mice injected with 6-OHDA and treated with GW842166x only experienced a small reduction in dopamine release (*p* < 0.001; Figure 3A,B), which was prevented by co-treatment with AM630 (*p* < 0.001; Figure 3A,B).

**Figure 3 biomedicines-10-01776-f003:**
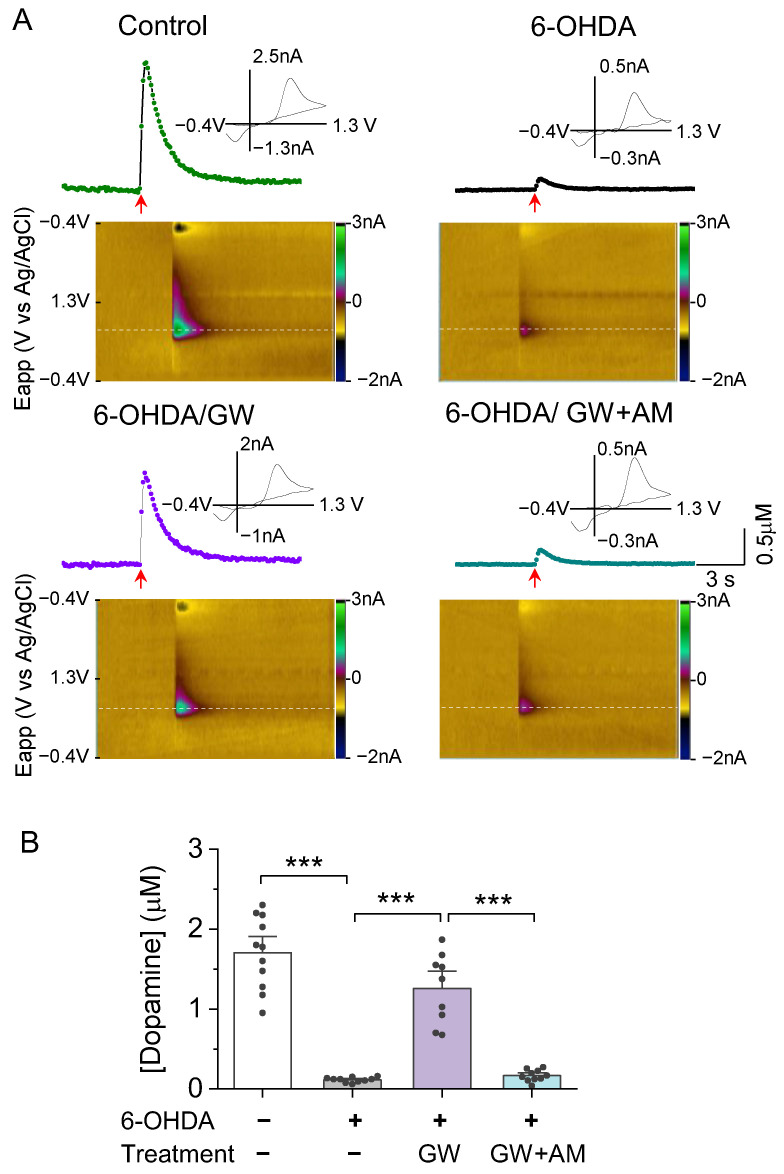
GW protected against 6OHDAinduced loss of evoked dopamine release in the NAc in vitro. (**A**) Concentration trace (top) and color plot (bottom) for dopamine release triggered by electrical stimulation of the NAc shell in slices from mice in the control and treatment groups. (**B**) Summarized data indicate that there was a significant decrease in evoked dopamine release in the 6-OHDA group relative to the control (*** *p* < 0.001; *n* = 10–11 slices from 3–4 mice). Chronic treatment with GW reduced 6-OHDA-induced loss of evoked dopamine release (*** *p* < 0.001; *n* = 9–10 slices from 3–3 mice), which was blocked by AM (*** *p* < 0.001; *n* = 9–10 slices from 3–3 mice).

We next determined whether the CB2 agonist affected dopamine dynamics in the striatum using in vivo fiber photometry to detect dopamine signals with the green fluorescent dopamine sensor GRAB_DA2m_ [53]. Mice received intra-striatal injections of 6-OHDA, followed by injection of a 1:1 mixture of AAV9-hSyn-GRAB_DA2m_ and AAV8-hSyn-mCherry. A 200 μm fiber-optic cannula was implanted at the same site followed by three weeks of drug treatments (Figure 4A). After three weeks to allow for GRAB_DA2m_ expression and drug treatments, fiber photometry was performed to detect striatal dopamine release (Figure 4B,C). A patch cable was attached to the implanted cannula and mice were placed into an open field chamber. Mice were allowed to habituate for 15 min before striatal dopamine events were recorded (Figure 4C). One-way ANOVA revealed a significant drug treatment effect on the frequency of dopamine events (*F*_3,23_ = 22.6, *p* < 0.001; Figure 4D). 6-OHDA induced a significant decrease in dopamine event frequency in the striatum compared to the control (*p* < 0.001; Figure 4D). GW842166x treatment attenuated the reduction in dopamine event frequency (*p* = 0.001; Figure 4D), which was prevented by co-treatment with AM630 (*p* = 0.032; Figure 4D). These results show that GW842166x was protective against 6-OHDA-induced reduction in striatal dopamine release via CB2 receptors.

**Figure 4 biomedicines-10-01776-f004:**
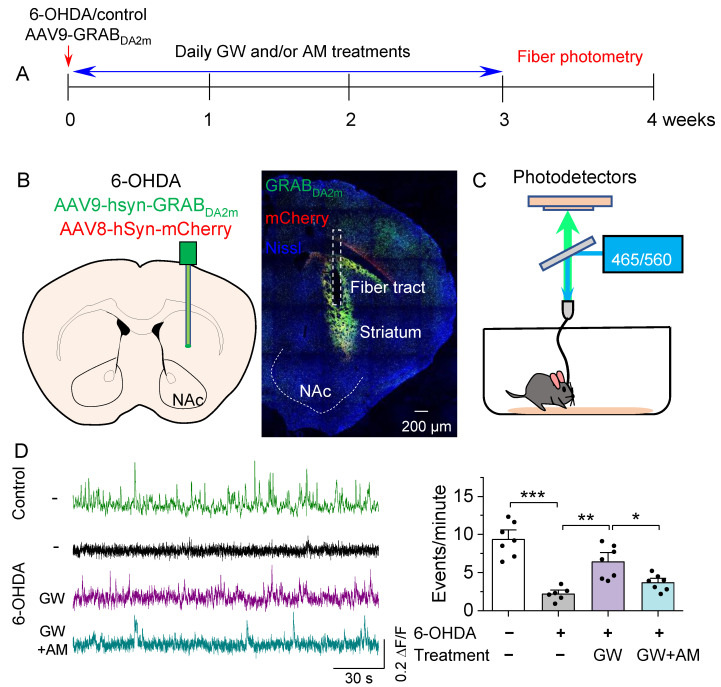
GW ameliorated the in vivo reduction in spontaneous dopamine transients in the NAc. (**A**) Timeline of intra-striatal injection of 6-OHDA, AAV9-GRABDA2m and AAV8-mCherry, drug treatments, and fiber photometry recording. (**B**) Diagram showing targeting of AAV and cannula implant to the NAc (left) and a representative post hoc validation of AAV expression and optic fiber position (right). (**C**) Schematic demonstrating the path of excitation LEDs and reporter emission signals during fiber photometry recording. (**D**) Left, representative traces of GRABDA2m ΔF/F in the NAc of mice in control and treatment groups. Intra-striatal 6-OHDA injections significantly decreased the number of events/min relative to control injections (*** *p* < 0.001; *n* = 6–7 mice). Right, chronic treatment with GW treatment mitigated the decrease in dopamine event frequency by 6-OHDA (** *p* = 0.001; *n* = 6–7 mice). The effect of GW was blocked by co-treatment with AM (* *p* = 0.032; *n* = 7–7 mice).

### 3.3. GW842166x Protected against 6-OHDA-Induced Anxiogenic- and Depressive-Like Behaviors

Anxiety and depression are the most common neuropsychiatric disorders in patients who have also been diagnosed with PD, with between 40 and 50% being diagnosed with comorbid depression [7]. 6-OHDA and other neurotoxic models of PD are associated with anxiogenic- and depressive-like behaviors in mouse models [38]. CB2 activation with GW842166x treatment protected against 6-OHDA-induced decreases in dopamine release both in vitro (Figure 3) and in vivo (Figure 4). We also previously showed that GW842166x-mediated CB2 activation protected against motor function deficits following intra-striatal injection of 6-OHDA [37]. Here, we investigated whether CB2 receptor agonists ameliorated anxiogenic- and depressive-like behaviors, the timeline of which is described in Figure 5A. Prior to behavioral testing, we examined whether body weight was impacted by 6-OHDA and/or drug treatment. One-way ANOVA showed that body weight was not significantly affected by 6-OHDA injection and drug treatment (*F*_3,31_ = 1.1, *p* = 0.370; Figure 5B).

The open field test (OFT) was used to assess whether 6-OHDA injection and CB2 activation altered anxiogenic-related behavior and locomotor activity, as the tendency of mice to avoid open spaces correlates with reduced time spent in the center of the open field chamber [54]. Mice injected with 6-OHDA exhibited a significant decrease in total distance traveled inside the field, indicating decreased motor function or anxiogenic-like behavior (*F*_3,31_ = 14.3, *p* < 0.001; Figure 5C). GW842166x treatment increased total distance traveled (*p* = 0.027; Figure 5C), and the effect was blocked by co-treatment of CB2 antagonist AM630 (*p* = 0.027; Figure 5C). Additionally, 6-OHDA significantly decreased the time spent in the center of the open field (*F*_3,31_ = 4.6, *p* = 0.009; Figure 5D), but neither GW842166x nor GW842166x + AM630 treatment significantly affected the time spent in the center (*p* > 0.05; Figure 5D).

Decreased sucrose preference in rodents has been interpreted as anhedonia [55], a core symptom of depression [56]. One-way ANOVA revealed GW842166x treatment significantly affected the sucrose preference score (*F*_3,31_ = 11.5, *p* < 0.001; Figure 5E). Tukey’s post hoc tests indicated that 6-OHDA-injected mice experienced significantly decreased sucrose preference compared to control mice (*p* < 0.001; Figure 5E), and this increase was attenuated by GX842166x treatment (*p* < 0.001; Figure 5E). The effect of GW842166x was prevented by co-treatment with AM630 (*p* = 0.040; Figure 5E). In the light–dark box test (LDT), less time spent in the light compartment and fewer transitions between compartments are associated with increased anxiogenic-like behavior [57,58]. Neither 6-OHDA injection nor drug treatments had a significant effect on these measures (light box time, *F*_3,31_ = 0.03, *p* = 0.994; compartment transitions, *F*_3,31_ = 1.6, *p* = 0.217; Figure 5F,G).

**Figure 5 biomedicines-10-01776-f005:**
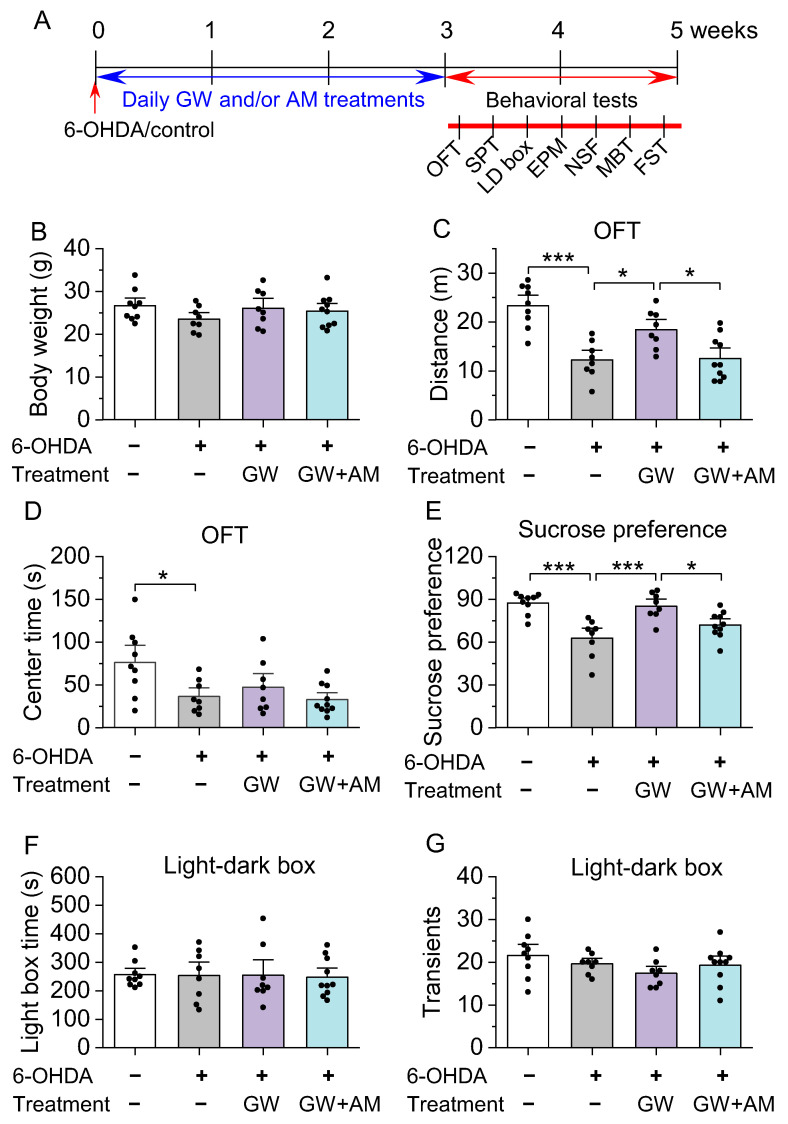
GW treatment attenuated 6-OHDA-induced deficits in the open field test (OFT), sucrose preference test (SPT), and light–dark box (LDB) test. (**A**) Timeline of 6-OHDA injection, drug treatments, and behavioral tests. (**B**) Neither 6-OHDA or drug treatments had a significant effect on body weight (*p* > 0.05). (**C**,**D**) In the OFT, 6-OHDA injection reduced total distance traveled ((**C**) *** *p* < 0.001; *n* = 8–9 mice) and time spent in the center of the chamber ((**D**) * *p* = 0.030; *n* = 8–9 mice) compared to the controls. GW treatment prevented the reduction in distance traveled (* *p* = 0.027; *n* = 8–8 mice), and the effect of GW was blocked by AM630 co-treatment (*p* = 0.027; *n* = 8–10 mice). However, neither GW alone (*p* = 0.867; *n* = 8–8 mice) or in combination with AM630 (*p* = 0.689; *n* = 8–10 mice) had a significant effect on time spent in the center of the chamber. (**E**) In the SPT, 6-OHDA injection significantly reduced sucrose preference (*** *p* < 0.001; *n* = 8–9 mice) relative to the control. GW treatment prevented the reduction in sucrose preference (*** *p* < 0.001; *n* = 8–8 mice), an effect that was blocked by AM630 (* *p* = 0.040; *n* = 8–10 mice). (**F**,**G**) In the LDB, neither 6-OHDA injection or treatment with GW and/or AM had a significant effect on time spent in the light compartment ((**F**) *p* > 0.05) or the number of transitions between the light and dark areas ((**G**) *p* > 0.05).

Similar to the OFT and LDT, a decrease in entries into and/or time spent in the open arms of the elevated plus-maze indicates anxiogenic-like behavior [59,60]. One-way ANOVA did not reveal a significant effect of drug treatment on the time spent in the open arms and the entries into the open arms (open arm time, *F*_3,31_ = 1.9, *p* = 0.145; open arm entries, *F*_3,31_ = 2.6, *p* = 0.068; Figure 6A,B). In the novelty-suppressed feeding test [61], mice were first deprived of food for 24 h. An increase in the latency to feed on food in the center of a novel open field is indicative of anxiogenic- and depressive-like behavior [61]. One-way ANOVA indicated that 6-OHDA-injected mice demonstrated a significant increase in the latency to feed in a novel environment compared to the control mice. The effect of drug treatment on the latency to feed was significant (*F*_3,31_ = 9.5, *p* < 0.001; Figure 6C). Tukey’s post hoc tests revealed that the 6-ODHA injection group showed a prolonged latency to feed (*p* < 0.001; Figure 6C) relative to the control mice, which was ameliorated by GW842166x treatments (*p* = 0.015; Figure 6C). The effect of GW842166x was blocked by co-treatment with AM630 (*p* = 0.049; Figure 5). However, there was no significant effect of drug treatment on the latency to feed in the home cage (*F*_3,31_ = 2.4, *p* = 0.087; Figure 6D).

Increased marble burying in the marble burying test reflects neophobic anxiety and is responsive to anxiolytics [62]. The 6-OHDA group buried significantly more marbles than the control mice (*F*_3,31_ = 8.9, *p* < 0.001; Figure 6E), indicating that 6-OHDA injection increased neophobic anxiety. GW842166x treatment significantly decreased the number of marbles buried following 6-OHDA injection (*p* = 0.002; Figure 6E), and this effect was blocked by co-treatment with AM630 (*p* = 0.037; Figure 6E). Immobility in the forced swim test is a measure of behavioral despair and reflects a depressive phenotype [63]. One-way ANOVA found the effect of GX842166x treatment in this test was significant (*F*_3,31_ = 13.0, *p* < 0.001; Figure 6F). Tukey’s post hoc tests revealed that 6-OHDA-injected mice showed significantly increased immobility time compared to the control mice (*p* < 0.001; Figure 6F), indicative of a depressive phenotype. The increase in immobility following 6-OHDA injection was attenuated by GX842166x treatment (*p* = 0.004; Figure 6F), and the effect of GW842166x was prevented by AM630 co-treatment (*p* = 0.035; Figure 6F). As examined above, 6-OHDA-treated mice exhibited anxiogenic- and depressive-like behaviors in 5 out of 7 behavioral tests. GX842166x treatments ameliorated anxiogenic- and depressive-like behaviors in a CB2 receptor-dependent manner.

**Figure 6 biomedicines-10-01776-f006:**
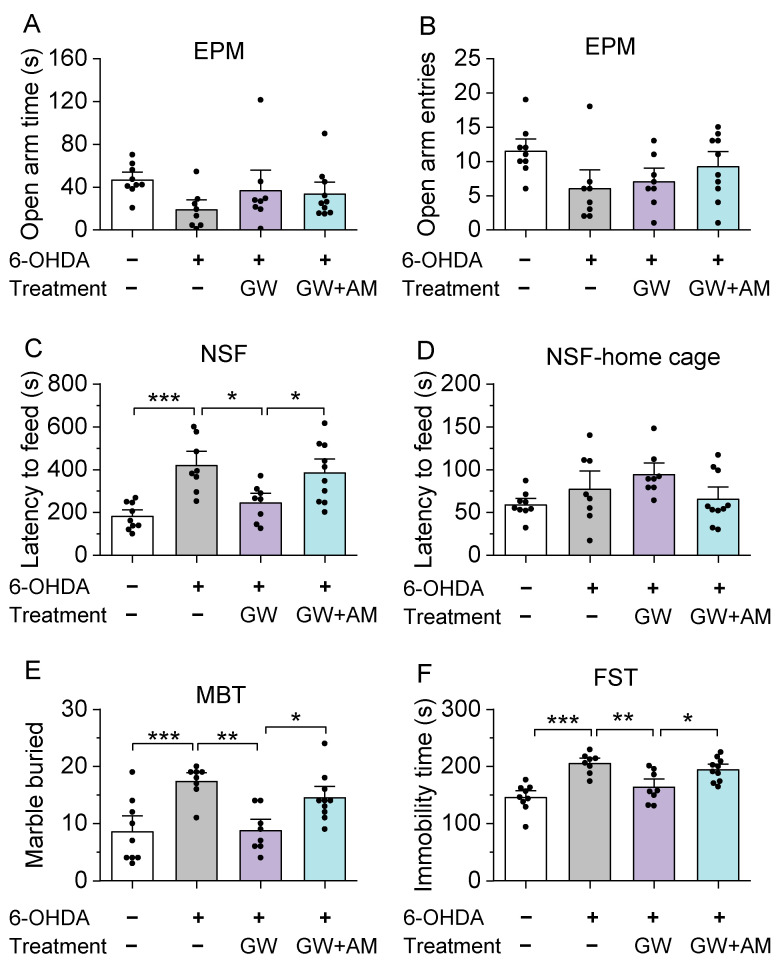
GW treatment attenuated 6-OHDA-induced deficits in the elevated plus-maze (EPM) test, novelty-suppressed feeding (NSF) test, marble burying test (MBT), and forced swim test (FST). (**A**,**B**) In the EPM test, neither 6-OHDA or drug treatments had a significant effect on open arm time ((**A**) *p* > 0.05) or the number of open arm entries ((**B**) *p* > 0.05). (**C**,**D**) In the NSF test, 6-OHDA injection increased feeding latency in a novel environment ((**C**) *** *p* < 0.001; *n* = 8–9 mice) compared to the controls. GW treatment prevented the increase in latency to feed in a novel environment (* *p* = 0.015; *n* = 8–8 mice), but co-treatment with AM blocked the effect of GW (* *p* = 0.049; *n* = 8–10 mice). However, when returned to the home cage (**D**) 6-OHDA injection and GW treatment had no effect (*p* > 0.05). (**E**) In the MBT, 6-OHDA injection increased the number of marbles buried (*** *p* < 0.001; *n* = 8–9 mice). Treatment with GW blocked the increase in marble burying (* *p* = 0.002; *n* = 8–8 mice), and co-treatment with AM prevented the effect of GW (* *p* = 0.037; *n* = 8–10 mice). (**F**) 6-OHDA injection increased immobility time in the FST (*** *p* < 0.001; *n* = 8–9 mice), which was ameliorated by treatment with GW (** *p* = 0.004; *n* = 8–8 mice). The effect of GW was blocked by AM co-treatment (* *p* = 0.035; *n* = 8–10 mice).

## 4. Discussion

Our prior work indicated that treatments with the selective CB2 agonist GW842166x protects against 6-OHDA-induced loss of dopamine neurons in the SNc and reduced motor function deficits in mouse models [37]. Here, we extended our previous study and further demonstrated the neuroprotective effects of GW842166x against 6-OHDA-induced reduction of dopaminergic axonal terminal density and dopamine release in the striatum. We found 6-OHDA-treated mice exhibited anxiogenic- and depressive-like behaviors, whereas GW842166x treatments ameliorated these behaviors. The two studies indicate that the CB2 agonist protects against 6-OHDA-induced neurodegeneration of dopamine neurons and associated deficits of motor and non-motor functions. The neuroprotective effects of GW842166x were blocked by the CB2 antagonist AM630, suggesting CB2-dependent mechanisms.

In our previous [37] and present studies, 6-OHDA was unilaterally injected into the dorsal striatum to induce dopamine neuron degeneration. Compared with 6-OHDA injection into the SNc and medial forebrain bundle, injection of 6-OHDA into the dorsal striatum causes partial lesions of SNc dopamine neurons that resemble earlier stages of PD, when neuroprotective agents exert major impact [64]. Consistent with previous studies [64], we found the loss of neurons was predominantly restricted to SNc dopamine neurons, as dopamine neurons in the VTA and non-dopamine SNc neurons were largely spared (Figure 1). 6-OHDA caused significant loss of TH^+^ dopamine axonal terminals in the striatum. Treatments with GW842166x for 3 weeks significantly attenuated the 6-OHDA-induced loss of dopaminergic axonal terminals in the striatum, and this effect was prevented by the selective CB2 antagonist AM630. Thus, this study provides further evidence that GW842166x protects against the degeneration of SNc dopamine neurons in a CB2 receptor-dependent manner.

FSCV recordings of dopamine dynamics in striatal slices provide a good measure of action potential-dependent dopamine release. Fiber photometry recordings of GRAB_DA2m_ signals in the striatum allow for real-time monitoring of dopamine transients in free-moving mice. These two approaches complement each other and provide a more complete picture of the change in dopamine release in the striatum. We found that 6-OHDA caused a drastic reduction in dopamine release in vitro and in vivo, GW842166x treatments ameliorated 6-OHDA-induced decreases in dopamine signals in the striatum, and the protective effects of GW842166x were blocked by the CB2 antagonist AM630. Regarding the potential mechanisms involved, we have previously shown that GW842166x decreased the action potential firing of SNc dopamine neurons, and this effect was likely mediated by a voltage-dependent decrease in the activation of the hyperpolarization-activated currents (I_h_) [37]. Ca^2+^ overload is an important factor that contributes to the degeneration of SNc dopamine neurons in PD [65]. 6-OHDA induces oxidative stress to dopamine neurons and Ca^2+^-mediated cytotoxicity [66]. The GW842166x-induced suppression of action potential firing likely reduces Ca^2+^ influx and prevents Ca^2+^ overload. These mechanisms may explain why GW842166x ameliorated the deleterious effects of 6-OHDA on dopamine release in the striatum. However, we cannot exclude the possibility that CB2 receptors in non-neuronal cells [67] may also confer the neuroprotective effects of GW842166x.

Depression is prevalent in PD patients with an occurrence rate of 40–50% [7,8,9]. Functional imaging studies have linked mesolimbic dopamine dysfunction to anxiety, depression, and apathy in PD patients [68,69,70]. Decreased dopamine release is associated with anxiogenic- and depressive-like behaviors [51]. We carried out a comprehensive array of behavioral tests to assess these behaviors. We found that 6-OHDA-treated mice exhibited anxiogenic- and depressive-like behaviors in the open-field, sucrose preference, novelty-suppressed feeding, marble burying, and forced swim tests but did not show significant change in the elevated plus-maze and light–dark box test. Although there is some overlap in anxiogenic- and depressive-like phenotypes, each is associated with a distinct neural pathway in resting-state functional connectivity MRI studies [71,72] of PD patients.

One caveat is that 6-OHDA-treated mice display motor function deficits [37], while many behavioral tests depend on proper motor functions. Would the impairment of motor functions alter the readout of these behavioral tests and lead to misinterpretation as anxiogenic- and depressive-like behaviors? Indeed, in the open field test, 6-OHDA-treated mice showed a decrease in total distance traveled and the time spent at the center of the open field chamber. It is possible that less time spent at the center could be attributed to an overall decrease in locomotor activity. The same argument could be applied to the forced swim test, where the increase in immobility in the 6-OHDA group could be attributed to impaired motor function. However, it is unlikely that impaired motor function contributed to the decrease in sucrose preference, as mice were equally amenable to both water and sucrose solutions. Anhedonia is a core symptom of depression [56], and the decreased sucrose preference suggests anhedonia-like behavior. In the novelty-suppressed feeding test, 6-OHDA-treatment led to an increase in latency to feed in the novel environment but not in the home cage, suggesting a depressive-like phenotype. In the marble burying test, 6-OHDA-treated mice buried more marbles, which could not be explained by impaired motor function. We have shown that dopamine release in the striatum was significantly decreased in 6-OHDA-treated mice. As dopamine is required for reward and motivation [73,74], a decrease in dopamine neuronal activity could lead to depressive-like phenotypes [75,76]. Together, these results suggest that 6-OHDA-treated mice exhibited anxiogenic- and depressive-like behaviors.

Notably, we found that GW842166x treatments ameliorated 6-OHDA-induced behavioral changes. GW842166x prevented decreased sucrose preference, increased marble burying, prolonged latency to feed, and increased immobility time in water. These effects of GW842166x were blocked by the CB2 receptor antagonist AM630. Mice with dopamine neuron-specific knockout of CB2 exhibited increased immobility time in the forced swim and tail suspension tests, suggesting a depressive-like phenotype. However, these mice were less aversive to the open arms of the elevated plus-maze and the light side of the light–dark box, suggesting anxiolytic-like behaviors [77]. Thus, CB2 receptors from midbrain dopamine neurons play an important and complex role in modulating affective behavior. With this caveat in mind, our results indicate that GW842166x treatments produce anxiolytic and antidepressant-like behaviors. The effects of the 6-OHDA lesion, across the range of behavioral measures tested, were prevented or attenuated following three weeks of GW842166x treatments in a CB2-dependent manner. As GW842166x was found to be safe in phase II clinical trials for chronic pain [36], it is our hope that CB2 agonists can be repurposed for neuroprotection and the mitigation of anxiety and depression symptoms in PD.

## Data Availability

The data presented in this study are available upon request from the corresponding author.

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
