# Peer review of "CB2 Agonist GW842166x Protected against 6-OHDA-Induced Anxiogenic- and Depressive-Related Behaviors in Mice"

_biomedicines, 2022, doi:10.3390/biomedicines10081776_

Round 1

Reviewer 1 Report

The manuscript entitled ”CB2 agonist GW842166x protected against 6-OHDA-induced anxiety- and depressive-related behaviors in mice” by Xiaojie Liu and collaborators is focusing on the potential role of GW842166x as treatment for neuropsychiatric symptoms in a Parkinson disease model induced by administration of 6-OHDA to C57BL/6J mice. The topic of the manuscript is highly relevant since the neuropsychiatric symptoms are slightly overlooked in the context of neurodegenerative disorders. Some minor concerns are listed below:

1     Throughout the manuscript, GW842166x is termed either treatment (lines 298, 353, 374, etc.) or pretreatment (lines 25, 301, 433, etc.). Is there a difference?

In the Materials and Methods section, please check the numbering.

Please mention the total number of animals used in this study.

Two anesthesia protocols are described in Materials and Methods section. I assume that ketamine and xylazine were used also for their analgesic properties, aside for their anesthetic ones. Is there an experimental reason for the use of isoflurane for Fast-scan cyclic voltammetry (FSCV)?

For section 2.3. Drug treatment, line 119 Group 2: Vehicle followed by daily i.p. saline injection, I assume that vehicle represents the injection of PBS and 0.02% sodium L-ascorbate into the striatum. However, at section 3.1 CB2 activation protected against the 6-OHDA-induced degeneration of dopamine axonal 276 terminals in the striatum, line 294, the term vehicle appears to represent the i.p. delivery of saline. If this is true, please establish the correct terminology and use it throughout the entire manuscript to avoid confusion.

In the legend of Figure 1, please mention the number of animals used.

As appears from the legend of Figure 2 to 6, the groups of animals had different sizes. Please explain.

At line 375, please correct deceases.

From line 428 to line 433, please check. The information provided seems to be contradictory.

In the Discussion section, lines 546 to 548, please add citation. 

-    

Author Response

We thank the reviewer for all the feedback and suggestions, and found that they have greatly improved the quality of the manuscript. We hope that the manuscript will be suitable for publication following these revisions.

1, Throughout the manuscript, GW842166x is termed either treatment (lines 298, 353, 374, etc.) or pretreatment (lines 25, 301, 433, etc.). Is there a difference?

Response: These two are the same. We have changed the term into treatment throughout the manuscript as GW was given after 6-OHDA.

2, In the Materials and Methods section, please check the numbering.

Response: Corrected.

3, Please mention the total number of animals used in this study.

Response: The number of animals used in each measurement has been added to the Materials and Methods section (section 2.1) and each figure legend.

4, Two anesthesia protocols are described in Materials and Methods section. I assume that ketamine and xylazine were used also for their analgesic properties, aside for their anesthetic ones. Is there an experimental reason for the use of isoflurane for Fast-scan cyclic voltammetry (FSCV)?

Response: The different anesthesia protocols were chosen based on the tradition of our lab. Ketamine/xylazine combination was used for survival surgery and transcardial perfusion. Isoflurane inhalation was used for acute brain slice preparation, which provides short-term anesthesia during decapitation.

5, For section 2.3. Drug treatment, line 119 Group 2: Vehicle followed by daily i.p. saline injection, I assume that vehicle represents the injection of PBS and 0.02% sodium L-ascorbate into the striatum. However, at section 3.1 CB2 activation protected against the 6-OHDA-induced degeneration of dopamine axonal 276 terminals in the striatum, line 294, the term vehicle appears to represent the i.p. delivery of saline. If this is true, please establish the correct terminology and use it throughout the entire manuscript to avoid confusion.

Response: Sorry for the confusion as we had control injections for 6-OHDA (in the striatum) and GW842166x treatment (i.p.). We changed intra-striatum injected solution (PBS and 0.02% sodium L-ascorbate) into “control”, and the i.p. injected solution into “vehicle”. All the terms have been made consistent throughout the manuscript.   

6, In the legend of Figure 1, please mention the number of animals used.

Response: We have added the number of animals used to the legend of Figure 1 (n = 3 mice). As we did the DAB immunostaining for TH in our paper recently published in Cells, we did not do a comprehensive study to qualify the changes in SNc dopamine neurons (to avoid repeating our Cells study).     

7, As appears from the legend of Figure 2 to 6, the groups of animals had different sizes. Please explain.

Response: Thanks for pointing this out. We have added the detailed description in the methods section. In most cases, we used different cohorts of animals for different experiments. Figure 2 and 3 used same cohort of animals (Timeline in Figure 2A). Fiber photometry in Figure 4 used a new cohort of mice (see timeline in Figure 4). Anxiogenic- and depressive-like behaviors in Figures 5 and 6 used the same new cohort of animals. Please see timeline in Figure 5A.

8, At line 375, please correct deceases.

Response: Corrected.

9, From line 428 to line 433, please check. The information provided seems to be contradictory.

Response: Thanks for pointing out. We have corrected this error.

10, In the Discussion section, lines 546 to 548, please add citation.

Response: Added.

Reviewer 2 Report

The present work by Liu et al., “CB2 agonist GW842166x protected against 6-OHDA-induced 2 anxiety- and depressive-related behaviors in mice” is interesting and well written. It addresses an area of Parkinson disease that has received less attention than the motor defect and evaluate a therapeutic avenue. The figures are clear and the inclusion of experimental schematics make the manuscript accessible to all reader and therefore accessible to the broad audience of this journal.

While the work is interesting, I have a few comments listed below:

Overall, how the behavior test results were analyzed is poorly described and more details should be included.

Methods:

- Inconsistency in the companies reporting: city, country, ... this should be addressed.

- The liquid food should be described.

- The rationale behind using young mice in a Parkinson Disease study have to be explained as it is usually associated with older individuals.

- The solvent of AM630 appears to be 35% PEG 200, therefore why only using saline as control?

- Please justify why the group: 6-OHDA followed by daily i.p. injections of 10 mg/kg AM630 was not included.

- The clone or the catalogue number of the antibodies have to be provided.

- ImageJ version and reference are missing. If any particular plugins were used these must be referenced too.

- More details regarding how the optical densities were measured have to be provided to be able to evaluate how the author analyzed their images.

- The Python code should be provided in supplemental data or deposited on an accessible repository.

- How were the z scores calculated?

- Were the behavior tests performed at similar time? This impact the reproducibility of the tests and therefore should be provided

- Why starving the animal for 24h for the NSF test knowing this may interfere with the test? https://www.ncbi.nlm.nih.gov/pmc/articles/PMC6408535/

- What software was used to run the statistics?

Results:

- Figures: the scale bars are missing.

- When comparing the change in body weight no differences are measured between groups including both males and females. Could differences be observed when comparing only males or females?

- Could the results of the open field test suggest an anxious behavior rather than a decreased motor function? As suggested in the discussion if the mice had a severely decreased motor function this would also affect the swimming test.

- The influence of gender in the different behavior should also be included in the different comparison especially in models used to mimic Parkinson disease known to affect men earlier and more frequently 10.1136/jnnp.2006.103788

- The authors suggest that neither GW842166x nor AM630 treatment significantly affected the time spent in the center. GW842166x nor GW842166x+AM630 treatments would be more accurate according to the results

Discussion:

-   A densitometric analysis of the TH+ and NeuN staining would help the discussion by providing numbers to refer to rather than a single image in the Figure 1

References:

- issues with the references 47 and 48

Author Response

We thank the reviewer for all the feedback and suggestions, and found that they have greatly improved the quality of the manuscript. We hope that the manuscript will be suitable for publication following these revisions.

-Overall, how the behavior test results were analyzed is poorly described and more details should be included.

Response: Thank you for pointing this out. The detailed statistical methods for analyzing behavioral test results have been described in Methods section. We have also added more details in Results section.

Methods:

- Inconsistency in the companies reporting: city, country, ... this should be addressed.

Response: Addressed.

- The liquid food should be described.

Response: Following surgery, mice in all groups of experiments were fed with liquid food 20% Ensure® Original Vanilla Nutrition Powder (Abbott Laboratories, Abbott Park, IL, USA). We have added this info into the manuscript. We found that this liquid food has increased the survival rate of the animals and helped the recovery.

- The rationale behind using young mice in a Parkinson Disease study have to be explained as it is usually associated with older individuals.

Response: The reviewer is correct in pointing out that Parkinson’s Disease (PD) occurs more often in seniors. Many genetic models of PD have a late onset of motor deficits. However, the current study employed a neurotoxin model (6-OHDA) in adult mice (10–12-week-old at the beginning of the experiments), which is less sensitive to age. Many other papers used the similar age.

- The solvent of AM630 appears to be 35% PEG 200, therefore why only using saline as control?

Response: In fact, we always matched the vehicle used for drug injection whenever possible. We have corrected this error by using “vehicle” instead.

- Please justify why the group: 6-OHDA followed by daily i.p. injections of 10 mg/kg AM630 was not included.

Response: We carefully considered this issue in the very beginning and did not include the AM630 group in our recently published Cells paper. We tried to use the minimal number of animals necessary to allow us to draw the most critical conclusions. We felt that the AM630 alone was not essential.

- The clone or the catalogue number of the antibodies have to be provided.

Response: Added.

- ImageJ version and reference are missing. If any particular plugins were used these must be referenced too.

Response: The version of ImageJ and the related references have been added to the manuscript.

- More details regarding how the optical densities were measured have to be provided to be able to evaluate how the author analyzed their images.

Response: Added.

- The Python code should be provided in supplemental data or deposited on an accessible repository.

Response: The Python code has been provided as supplemental file.

- How were the z scores calculated?

Response: The z scores were calculated as described in the Materials and Methods: “A linear fit was applied to the data and the fitted 560 nm signal was aligned and subtracted from the 465 nm signal, and then divided by the fitted 560 nm signal to calculate the ΔF/F values over time.”

- Were the behavior tests performed at similar time? This impact the reproducibility of the tests and therefore should be provided

Response: Thank you for noting this omission. Beginning the day after completion of drug treatments, mice were subjected to a battery of tests of anxiety- anxiogenic- and depressive-like behaviors. We have clarified to note that tests were performed between 9:00 AM-12:00 PM each day.

- Why starving the animal for 24h for the NSF test knowing this may interfere with the test? https://www.ncbi.nlm.nih.gov/pmc/articles/PMC6408535/

Response: Food deprivation (drinking water was available) for 24 hours is widely used for the NSF test in the literature. In the NSF test, a fasted mouse faces the conflicting choice between eating and avoiding a potentially dangerous novel environment. Without prior fasting, the mouse does not have enough motivation to eat, and we therefore chose a 24 hour fast. All 4 experiment groups underwent the same fasting, and only two tests – marble-burying and forced-swim – were performed after NSF. Therefore, we believe that fasting should not interfere with the test.   

- What software was used to run the statistics?

Response: Origin2022b was used to run all the data statistics. We have this added to the Materials and Methods section.

Results:

- Figures: the scale bars are missing.

Response: Added.

- When comparing the change in body weight no differences are measured between groups including both males and females. Could differences be observed when comparing only males or females?

Response: We did not see significant differences in body weight between different treatment groups when comparing only males or females. Roughly equal number of male and female mice was used for all experiment groups.

- Could the results of the open field test suggest an anxious behavior rather than a decreased motor function? As suggested in the discussion if the mice had a severely decreased motor function this would also affect the swimming test.

Response: It is possible. However, it is difficult to draw firm conclusions from open field test and the forced swimming test as impaired motor function may confound data interpretation, as mentioned in Discussion.

- The influence of gender in the different behavior should also be included in the different comparison especially in models used to mimic Parkinson disease known to affect men earlier and more frequently 10.1136/jnnp.2006.103788

Response: Approximately equal numbers of male and female mice were utilized in all experimental groups. Except bodyweight, no significant sex differences were observed for any measures, thus the data were pooled across sexes in all studies. 

- The authors suggest that neither GW842166x nor AM630 treatment significantly affected the time spent in the center. GW842166x nor GW842166x+AM630 treatments would be more accurate according to the results

Response: We have revised the statement by following the reviewer’s suggestion.

Discussion:

-   A densitometric analysis of the TH+ and NeuN staining would help the discussion by providing numbers to refer to rather than a single image in the Figure 1

Response: As we did the DAB immunostaining for TH in our paper recently published in Cells, we did not do a comprehensive study to quantify the changes in SNc dopamine neurons (to avoid repeating our Cells study).     

References:

- issues with the references 47 and 48

Response: The reference errors have been fixed.
